# Engineering of phenylalanine dehydrogenase from *Thermoactinomyces intermedius* for the production of a novel homoglutamate

Muhammad Tariq[1,2], Muhammad Israr [1]*, Muslim Raza[3]*, Bashir Ahmad[1], Azizullah Azizullah[1], Shafiq Ur Rehman[1], Muhammad Faheem[4], Xinxiao Sun[2], Qipeng Yuan[2]*

**1** Department of Biology, The University of Haripur, Haripur, Khyber Pakhtunkhwa, Pakistan, **2** School of Life Science, Beijing University of Chemical Technology, Beijing, China, **3** Department of Chemistry, Bacha Khan University, Charsadda, Khyber Pakhtunkhwa, Pakistan, **4** Department of Biological Sciences, National University of Medical Sciences, The Mall, Rawalpindi, Pakistan

\* m.israr@uoh.edu.pk (MI); dr.raza@bkuc.edu.pk (MR); m.israr88@hebtu.edu.cn (QY)

**Data Availability Statement:** All relevant data are within the paper and its Supporting information files.

## Abstract

The dramatic increase in healthcare costs has become a significant burden to this era. Many patients are unable to access medication because of the high price of drugs. Genetic engineering has made advances to increase the yield, titer, and productivity in the bio-based production of chemicals, materials of interest, and identification of innovative targets for drug discovery. Currently, the production of homoglutamate (α-Aminoadipic acid) involves petrochemical routes that are costly with low yield and often not suitable for industrial production. Here, we established the development of NADH-dependent homoglutamate by engineering NADH-dependent phenylalanine dehydrogenase (PDH) from *Thermoactinomyces intermedius*, which provides a novel tool for in-vivo metabolic engineering and in-vitro catalysis. Based on computational insight into the structure, we proposed the site-specific directed mutagenesis of the two important residues of PDH through docking simulations by AutoDock Vina which elucidated the binding mode of PDH with α-Ketoadipic acid and ligands. Our results demonstrated that the catalytic efficiency $K_m/K_{cat}$ of the final mutant Ala135Arg showed a 3-fold increase amination activity towards the ketoadipic acid as compared to the other mutant Gly114Arg, a double mutant Gly114Arg/Ala135Arg, and wild type *TiPDH*. Furthermore, we have introduced formate dehydrogenase as a cofactor regenerative system in this study which further made this study economically viable. Our study unfolds the possibility of biosynthesis of other non-proteinogenic amino acids that might be valuable pharmaceutical intermediaries.

## Introduction

α-Aminoadipic acid also known as homoglutamate is a non-proteinogenic amino acid. It is a well-known acyclic precursor of the penicillins and cephalosporins antibiotics [1]. It is a potent prognosticator of prostate cancer [2], a biomarker for diabetes [3], a pharmacological tool as a

**Funding:** The author(s) received no specific funding for this work.

**Competing interests:** The authors have declared that no competing interests exist.

notable inhibitory of L-glutamate uptake, and a hypothetically precarious phase in the recycling of neurotransmitter glutamate [4]. It is reported as an important intermediary metabolite in the α-aminoadipate pathway during the metabolism of lysine and saccharopine and penicillin in β-lactam-producing *fungi* [5]. It has also been observed as an intermediary metabolite in the catabolism of L-lysine in mammals and β-lactam-producing filamentous fungi [6]. In addition, this non-proteinogenic amino acid can be synthesized through different petro-chemical routes, which are costly due to expensive raw materials with low yield and are not suitable for industrial production [7, 8]. Growing concerns over the formation of toxic byproducts and high cost have inspired a quest to provide a biosynthetic route for α-aminoadipic acid that makes efficient use of inexpensive raw materials, renewable resources and minimizes the formation of toxic byproducts [9]. Thus metabolic engineering has made advances to increase the yield, titer, and productivity in the bio-based production of chemicals, materials of interest, and identification of innovative targets for drug invention.

In β-lactam producing bacteria, L-lysine is directly converted into 2-aminoadipic acid for cephamycin biosynthesis [7, 10]. In this case, the amino group on C-6 of the donor lysine molecule is trans-aminated to an accepter 2-ketoglutarate or another 2-ketoacid. The resulting 2-aminoadipic semialdehyde is consequently oxidized to 2-aminoadipic acid [11]. In the case of *Penicillium chrysogenum*, for the conversion of L-lysine into α-aminoadipic acid two different pathways are involved, the one pathway in which lysine 2-ketoglutarate reductase results in the formation of saccharopine from lysine by extracting one amino group to be used as a nitrogen source. While in the second pathway the ω-aminotransferase converts D-lysine or L-lysine into 2-aminoadipic semialdehyde and 2-aminoadipic acid (homoglutamic acid). Both enzymes are prompted by L-lysine which is negatively regulated by $NH_4^+$ ions. The ω-aminotransferase from different species uses 2-ketoacids as amino group acceptors. The most common ones are 2-ketoadipate and 2-ketoglutarate resulting directly in the development of 2-aminoadipic acid and glutamic acid [11–13]. The pyruvate also has been reported as an amino acceptor in ω-aminotransferase from *Pseudomonas putida* and *Pichia guillermondii* catalyzed reactions [14, 15].

The phenylalanine dehydrogenase (PDH) enzyme was first reported from *Brevibacterium sp*. that catalyze the reversible deamination of L-phenylalanine to phenylpyruvate and ammonia in the presence of NAD [16]. It is also significant for the production of L-phenylalanine and its associated L-aminoacids from the corresponding Oxo-analogues and ammonia [17].

This enzyme from *Bacillus badius* was reported for more specific identification of phenylketonuria (PKU) disease [18]. The phenylalanine dehydrogenase from *Thermoactinomyces intermedius* is pro-S stereospecific for hydrogen transfer of NADH and has high substrate specificity and thermostability for amino acid biosynthesis [16]. It inherits hexapeptide segments (124F-V-H-A-A-129R) in the substrate-binding domain and is responsible for directing the most desirable substrates of the phenylpyruvate, phenyl-alanine, and phenylalanine dehydrogenase [19]. Comparison of different phenylalanine dehydrogenases has shown substantial resemblance particularly in the region between D58-E130. The region from D58 to E130 contains a 78G-G-G-81K domain, which is common in different aminoacid dehydrogenases and probably constitutes the catalytic domain [17].

In the present study, we have chosen reductive amination reaction strategies for the production of homoglutamate which is an interesting route for the conversion of 2-ketoadipic acid catalyzed by NADH-dependent phenylalanine-dehydrogenase. To our knowledge, we have reported for the first time the engineering of phenylalanine dehydrogenase enzyme in *E. coli* for the production of homoglutamate as shown in Scheme 1. Previously, the affinity of TiPDH towards the transamination of 2-ketoadipic acid as substrate was found insufficient. Therefore, based on computational insight into the structure of TiPDH, we proposed the site-

specific directed mutagenesis of the two important residues of PDH through docking simulations by AutoDock Vina. Subsequently, it elucidated the binding mode of PDH with α-Ketoadipic acid and ligands by mutating the two important residues in the active sites of TiPDH that resulted in the production of homoglutamate. We have changed Gly114 to Arg114 and Ala135 to Arg135. The Ala135/Arg135 presented substantial performance compared to the wild-type enzyme and displayed about 3 fold increases towards the substrate. Besides we used formate dehydrogenase (FDH) in our system as a cofactor regenerative enzyme. The equilibrium of the system with co-factor regeneration abundantly considered homoglutamic acid production and $CO_2$ as the reaction byproduct. In this case, the biocatalysts used in the reductive amination process are favorably retained to advance the sustainability of the reaction system.

**Scheme 1. Biosynthesis of α-aminoadipic acid pathway**.

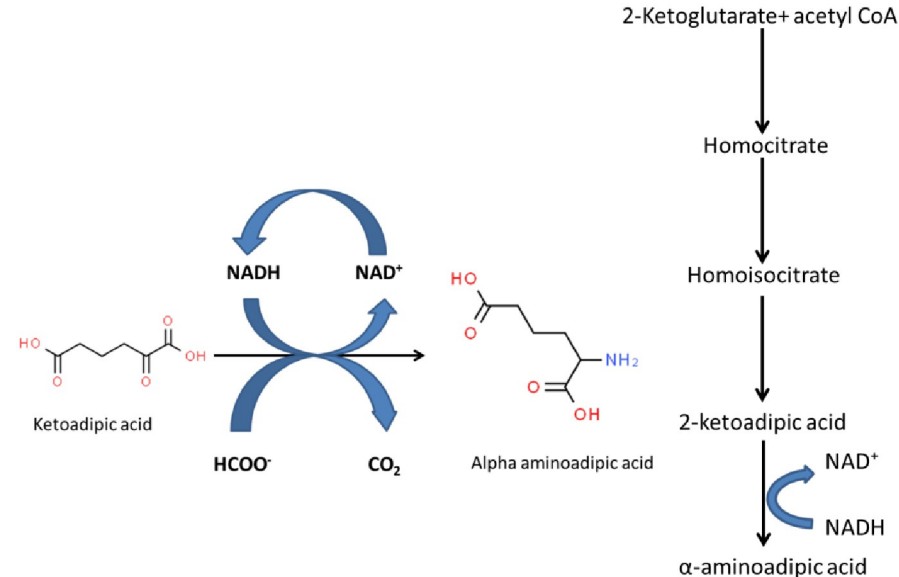

## Materials and methods

### Bacterial strains, plasmids, and culture conditions

Bacterial strains and plasmids used in this study are listed in S1 Table. The pET-Duet1 backbone was used for gene cloning and plasmid construction, whereas the *Escherichia coli (E. coli) BL21-DE3* strain was used for protein expression and homoglutamate production. Genomic DNA was isolated through (Invitrogen DNA Mini Kit according to manufacturer instructions) from different bacterial strains. All the primers used in this study are scheduled in S2 Table. The Luria-Bertani (LB) medium containing 10 g/L tryptone, 10 g/L NaCl, and 5 g/L yeast extract were used for inoculant preparation, protein overexpression, and cells propagation. The modified M9 medium contained M9 medium glucose 5 g/L, MOPS 2 g/L, glycerol 10 g/L, disodium hydrogen-phosphate 6 g/L, potassium dihydrogen-phosphate 3 g/L, ammonium chloride 1 g/L, sodium chloride 0.5 g/L, yeast powder 2 g/L was used for homoglutamate production. The *BL21-DE3* strain was cultivated at 30°C and a suitable amount of kanamycin 50 μg/mL, ampicillin 100 μg/mL, and chloramphenicol 34 μg/mL were added to the medium when needed.

## Gene cloning and mutagenesis

The His-tagged gene encoding phenylalanine dehydrogenase from *Thermoactinomyces interme-dius (TiPDH)* (Accession # D00631), *Sporosarcina ureae (SuPDH)* (Accession # AB001031), and other genes encoding *leucine dehydrogenase* (LDH) (Accession # CAA55671), *glutamate dehy-drogenase* (GDH) (Accession # BAD69594), *meso-diaminopimelate dehydrogenase* (DAPDH) (Accession # BAD40410) and *formate dehydrogenase* (FDH) (Accession # AIY34662) from *Ther-moactinomyces intermedius, Bacillus Subtilis, Symbiobacterium thermophilum,* and *Candida boi-dinii* respectively were augmented by PCR from extracted genomic DNA using prime star DNA polymerase. PCR was performed under the conditions, 30 cycles for each gene, each comprising denaturation at 96°C for 1 min, 30 sec for annealing at 58°C, and polymerized for 10 min at 72°C. Briefly, genes for TiPDH, SuPDH, and *Bacillus Subtilis* glutamate dehydrogenase (*BSGDH*) were digested via restriction sites *Bam*H1 and *Xho*l and *DAPDH* and *FDH* using *Bam*H1 and *Kpn*l while TiPDH using *Eco*R1 and *Xho*l and subcloned into plasmid *pET-Duet1*. The resulting plasmids were then transformed into chemically competent *E. coli* strain *Trans 5α*.

For mutagenesis two single mutations Gly114Arg and Ala135Arg were constructed through "Splicing by overlap extension SoE PCR" using Taq DNA Polymerase in the TiPDH gene. The Ala codons GCG were replaced by CGC (Arg) and Gly codon GGC by CGC (Arg). The SoE PCR conditions were set, a total of 25 cycles, each containing denaturation at 94°C for 30 sec, annealing at 62°C for 30 sec, and polymerization at 72°C for 7 min by using for both rounds of SoE PCR. During the 1st round, the external primers were used in combination with a set of complementary pairs of oligonucleotides, which are listed in S2 Table containing the mutated codons. The resulting overlapping fragments were then purified through the gel and re-com-bined in the 2nd round overlap extension PCR. The resulting amplicons were digested with *Eco*R1/*Xho*l, gel purified, and then ligated at 22°C into *pET Duet1* expression vector with T4-DNA ligase for 2 h. Afterward, the constructs were then confirmed by sequencing the entire open reading frame and were transformed into chemically competent *E. coli BL21* (*DE3*) cells (Stratagene) for protein expression.

## Protein expression and purification

For protein expression, a fresh colony of *BL21 (DE3)* with the recombinant plasmid pET-Duet1 containing His-tagged wild type or mutant gene was inoculated into a 4 mL LB media tube having ampicillin 100 μg/mL and incubated at 37°C overnight in the shaker. Then 0.5 mL overnight culture was inoculated to 100 mL LB medium containing ampicillin 100 μg/mL and incubated at 37°C in a shaker with 220 rpm. After reaching the optical density (OD$_{600}$) to 0.5, culture was induced through 0.5 mM isopropyl β-D-1-thiogalactopyranoside (IPTG) for pro-tein expression and incubated with shaking at 37°C for the next 12 h. The fermented cells were harvested at a high-speed centrifuge at 6000 rpm for 10 min. The harvested cells were then washed with the lysis buffer and disrupted by sonication (3-sec burst with 5-sec break) for 20 min. The disrupted cell debris was separated by centrifuging at 6000 rpm for 30 min at 4°C. The resulting clean supernatant was loaded to the Ni-NTA column and the column was washed by 30, 50, 300, and 500 mM of imidazole solution respectively. The eluted samples were analyzed with 12% (w/v) SDS-PAGE (sodium dodecyl sulfate-polyacrylamide gel electro-phoresis) whereas the protein concentration was then determined using the BCA kit according to the protein standard curve.

## Preparation of crude extracts and enzyme assays

All assays were performed in triplicate at 37°C. To test the transaminase activity, the genes TiLDH, TiPDH, SuPDH, BsGDH, and DAPDH were cloned into plasmid pETduet-1,

respectively and protein expression and purification were performed. The purified protein was used for in vitro activity assay. The decrease in NADH (nicotinamide adenine dinucleotide) absorption at 340 nm in reaction system was found under the following condition: 1 mL reaction mixture containing 100 mM Tris-NH$_4$Cl buffer (pH = 8.0), 1 mM NADH, 200 μL enzyme solution and 1 mM substrate 2-ketoadipate. During the reactions reduced NADH was converted into NAD$^+$ and absorption 340 nm was measured by ultraviolet spectrophotometer.

## Homology modeling of phenylalanine dehydrogenase

The target sequence of L-phenylalanine dehydrogenase *Thermoactinomyces intermedius* (TiPDH) was retrieved from the UniProt database [20]. The protein structure of L-PDH from the *Rhodococcus sp* (PDB code: 1C1D) was designated as the best model for the TiPDH homology model building using the bioinformatics tool the "BLAST online tool" [21, 22]. The template 1C1D showed 31% sequence similarity with the target protein having the highest resolution of 1.25 Å as compared to other available structures in the PDB. Bioedit software was used for the alignment of the target sequence with the template [23]. The 3-D coordinates of the template 1C1D and alignment file were used for the construction of homology models by using MODELLER 9.12 [24]. The Swiss PDB viewer v4.1.0 was used to refine the models, [25] while GROMOS 96 force field in the Swiss PDB Viewer was implemented to refine every residue. Energy minimization was performed (500 steps of steepest descent followed by 1000 steps of the conjugate gradient) without conveying any limitation. All the residues assume a stable conformation by avoiding steric interference [26]. Several other tools were used to validate the predicted models. The stereochemical quality of the protein like geometry checks, symmetry checks, (bond lengths, chirality, torsion angles, bond angles, etc.) was checked by using the PROCHECK online tool [27]. ProSA online server and ProSA2003 [28] tool was utilized to check the fitness of sequence to structure and validation of predicted 3-D models.

## Molecular docking simulations

To elucidate the binding mode of TiPDH with α-Ketoadipic acid, in silico docking simulations and substrate docking methods were carried out through AutoDock Vina software [29]. The finest model TiPDH 3-D structure was selected, visualized, and the solvent molecules from protein macro-molecule were removed through different evaluation tools as discussed. The compound α-Ketoadipic acid and phenylalanine ligands structures were prepared through both Avogadro's and Chem sketch Softwares [30, 31]. PyRex tool was connected with Auto-Dock Vina [32]. The calculation of gasteiger charges and the addition of hydrogen to α-Ketoadipic acid and phenylalanine were performed. All rotatable bonds of molecules were defined by default and were allowed to rotate during the automated docking process. Both the small molecule structures and prepared protein were used to calculate energy grid maps. A grid box size of 65× 65 × 65 Å points with a grid spacing of 0.8 Å was considered focusing on the center. Various binding affinities and frequent clusters were found for α-Ketoadipic acid and the best conformers were selected due to the top-ranked cluster and lower docked free energy to perform docking analysis with the LIGPLOT$^+$ version v.1.4.5 [33] Discovery studio visualizer version 4.0 [34], and PyMOL version 1.7.2 [35]. For authentication, the docking method was optimized through ligand (L-Phenylalanine) of the enzyme phenylalanine dehydrogenase. The structure of (L-Phenylalanine) was extracted from the binding pocket and re-docked to the TiPDH macromolecule. Two important residues (Ala135Arg and Gly114Arg) were mutated and all the docking procedure was again repeated for the macromolecule.

## Microbial production of homoglutamate with 2-ketoacid

The *E. coli* strain *BL-21* was transformed with different pET-Duet plasmids harboring both wild and mutant TiPDH. A single colony of producing strain was picked and cultivated in 4 mL LB medium with the appropriate antibiotic and grown overnight at 37˚C. Subsequently, the overnight pre-inoculum was transferred to a 50 mL modified M9 medium containing an appropriate amount of antibiotics and grown for 3 h at 37˚C with vigorous shaking (220 rpm). The cells were then supplemented by 1 mM 2-ketoadipate as a substrate and enzyme expression was induced by adding 0.5mM IPTG after reaching the $OD_{600}$ to 0.5 and then transferred into shakers at 30˚C for cultivation. The samples were taken after 12 h, 24 h, and 48 h, and cell growth was monitored by measuring the $OD_{600}$. The supernatant was centrifuged at (12000 rpm for 2 min) to remove the insoluble materials. The 200 μL were filtered and then analyzed by HPLC (high-performance liquid chromatography) after the process of derivatization. Triplicate transformants were used for feeding experiments.

## High-performance liquid chromatography HPLC analysis

The homoglutamate as a non-natural amino acid has no obvious ultraviolet absorption. It is mostly evaluated by the amino acid analyzer and high-performance liquid-phase derivatization. The HPLC (Shimadzu) L-3530 is equipped with a reverse Diamonsil C18 column, and a UV light detector known as an evaporative light scattering detector (ELSD) as a detector to detect the product homoglutamic acid. The mobile phase contains solvent A: pure methanol (chromatographic grade) and solvent B: 1% trifluoroacetic acid with a flow rate of 0.5 mL/min. The column temperature was set at 40˚C, gas pressure 3.5 MPa, homoglutamic acid detection method using isocratic elution, the proportion of mobile phase B is 95% [36].

# Results

## Establishment of the recombinant strains

Five transaminase enzymes TiLDH, SuPDH, TiPDH, BsGDH, and DAPDH from different bacterial strains were screened according to the structural similarity of the substrate 2-ketoadipate for the production of Homoglutamate. These enzymes were then expressed in the pET-Duet1 plasmid and purified against the substrate for its best activity. The recombinant plasmid pET-duet1 containing the TiLDH, SuPDH, TiPDH, BsGDH, and DAPDH respectively was then transferred into *E. coli BL21-DE3* and named as pETduet-1-TiLDH, pETduet-1-SuPDH, pETduet-1-TiDH, pETduet-1-BS GDH, and pETduet-1-DAPDH.

## Substrate specificity

To investigate the catalytic activity of the five transaminase enzymes TiLDH, SuPDH, TiPDH, BsGDH, and DAPDH, we tested the in vitro activity of all these enzymes for the substrate 2-ketoadipic acid. The recombinant strains were cultured in the M9 medium and only the TiPDH and BsGDH showed affinity towards the substrate. The reaction system was 1 mL of Tris-NH$_4$Cl buffer (pH = 8.0) containing 1 mM of NADH, 200 μL of enzyme solution, and 1 mM of substrate 2-ketoadipic acid. During the reaction, NADH was converted to NAD$^+$ and NADH had maximum absorption at 340 nm. Only the phenylalanine dehydrogenase TiPDH from the *actinomycetes* and glutamate dehydrogenase BsGDH from *Bacillus Subtilis* showed the specific activity towards the substrate. The TiPDH showed the highest specific activity of 0.125 μmol/min/mg of protein as compared to the BsGDH which showed 0.008 μmol/min/mg of protein, whereas, the other recombinant strains did not show any affinity towards the substrate shown in Table 1.

Table 1. Substrate specificity of dehydrogenases.

| Species | Gene | Specific activity (μmol/min/mg protein) |
|---------|------|------------------------------------------|
| *Thermoactinomyces intermedius* | TiPDH | 0.125 |
| *Bacillus subtilis* | BsGDH | 0.008 |
| *Thermoactinomyces intermedius* | TiLDH | 0 |
| *Symbiobacterium thermophilum* | DAPDH | 0 |
| *Sporosarcina ureae* | SuPDH | 0 |

## Engineered TiPDH showed increased $K_{cat}/K_m$ values on NADH

We found that the activity level of phenylalanine dehydrogenase towards the transamination of 2-ketoadipic acid was insufficient, so we first hypothesized a model and combined the substrate and enzyme by using computer modeling. To get an insight into the mutated residues Gly114/Arg114 and Ala135/Arg135 and the binding pocket of 2-ketoadipic acid, we performed a computational analysis of the enzyme. The wild-type and mutant polypeptides were constructed in the pET Duet-1 vector and then transformed into *E. coli BL21-DE3* which showed identical mobilities on SDS/PAGE, demonstrating the actual mol. weight of 40.5 kDa. Thus the replacement of residues in the mutant enzymes consequently didn't dislocate the intrinsic oligomeric structure.

The enzyme-substrate affinity ($K_m$) value was deliberated for both wild-type TiPDH and mutated TiPDHs *Thermoactinomyces intermedius* at a fixed concentration of the substrate 2-ketoadipic acid after drawing Line weaver–Burk plot. The $K_m$ value of the wild-type TiPDH for the substrate was calculated to be 1.771 μM and $K_{cat}$ 0.308 min$^{-1}$ shown in Fig 1a whereas, for the mutated TiPDH the $K_m$ was 7.518 μM and the $K_{cat}$ value was 1.37 min$^{-1}$ shown in Fig 1b.

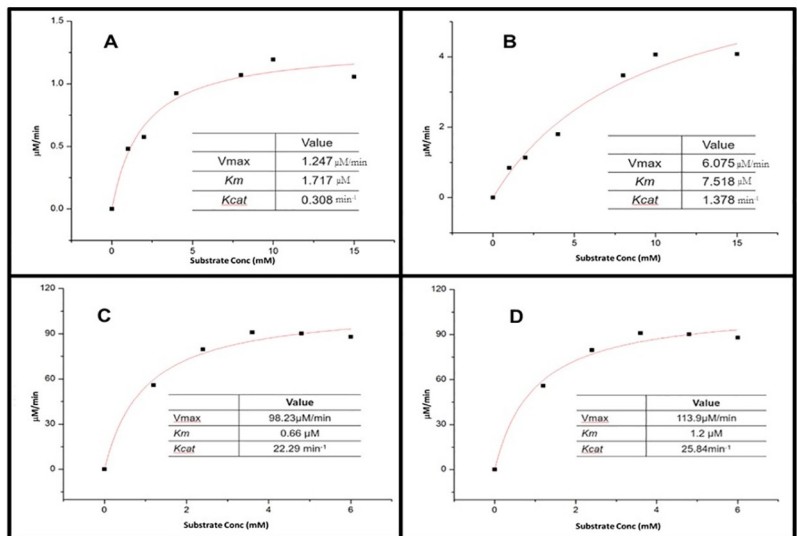

**Fig 1.** Enzyme substrate affinity of both mutant type and wild TiPDH (A) $K_m$ and $K_{cat}$ values of wild type TiPDH for the substrate 2-ketoadipic acid (B) $K_m$ and $K_{cat}$ values of mutant type TiPDH for the substrate 2-ketoadipic acid (C) $K_m$ and $K_{cat}$ values of wild type TiPDH for the natural substrate Phenyl Pyruvate (D) $K_m$ and $K_{cat}$ values of mutant TiPDH for the substrate natural substrate Phenyl Pyruvate.

The mutagenesis was in the manifestation of mutant Ala135Arg that displayed substantial alteration compared to the wild-type enzyme. The mutant Ala135Arg showed a significant amination activity towards the substrate, which was about 3-fold high as compared to wild TiPDH, indicating that the result of site-directed mutagenesis was positive. To study the changes in the catalytic properties of the enzyme after site-directed mutagenesis, the difference between wild TiPDH and mutant type TiPDH in the catalytic optimal substrate was first studied. It was found that the natural substrate for TiPDH was phenylpyruvate. We measured the specific activity of both mutant and wild-type TiPDHs towards the natural substrate. To illustrate the enzyme activity, both the mutant and wild-type TiPDHs were expressed and purified. The kinetic considerations for activation of 2-ketoadipic acid and its native substrate phenylpyruvate were determined by checking the consumption of NADH at 340 nm. The wild type TiPDH has a $K_{cat}$ value of 22.29 min$^{-1}$ and $K_m$ value of 0.66 μM as shown in Fig 1c for its natural substrate phenylpyruvate, whereas, the $K_m$ value of the mutant TiPDH for phenylpyruvate was 1.2 μM and the $K_{cat}$ value was 25.84 min$^{-1}$ shown in Fig 1d.

Phenylpyruvate was converted to phenylalanine by transaminase TiPDH and NADH. The $K_m$ and $K_{cat}$ values of the enzyme were increased but the affinity with the substrate decreased. The mutation sites convert the hydrophobic amino acids to hydrophilic amino acids, which results in a decrease in the binding affinity of the phenylpyruvate with the benzene ring of the enzyme. Based on protein structure analysis for the active site of phenylalanine dehydrogenase, an amino group of K78 hydrogen bonds to the carboxyl group of the substrate and determines the substrate specificity [37]. The mutation of alanine 135 to arginine increases the hydrophobic character hydrogen bonding of the binding pocket, which clarifies why specificity constant $K_m/K_{cat}$ mutant TiPDH towards 2-ketoadipic acid was 3-folds increased than that towards the native substrate phenylalanine. The Ala135Arg mutation increased the specificity constant ($K_{cat}/K_m$) toward the substrate in comparison with wild type and the other mutants Phenylalanine dehydrogenase, while for Gly114Arg and double mutant Gly114Arg/Ala135Arg showed no specific activity and were inactive toward the substrate.

## Optimizing the biosynthetic pathway of L-homoglutamate

The pH has a great influence on the state of ionization of basic or acidic amino acids which can lead to reformed protein recognition or an enzyme might become inactive. The alterations in pH may not only affect the shape of an enzyme but may also alter the shape or charge properties of the substrate so that either the substrate cannot bind to the active site or it cannot undergo catalysis. The optimum pH of the native enzyme TiPDH was found to be 8.4 and the temperature was 35°C. The reaction of the enzyme was allowed for 4 h at different temperatures in triplicate. The product was then tested for homoglutamate by HPLC and the conversion rate was calculated which showed that the optimum pH for transamination of 2-ketoadipic acid was 40°C and may affect the activity of transamination even at lower and higher pH. The effect of pH on the transamination reaction is shown in Fig 2a.

Similarly, the enzymatic activity remained high at optimum temperature, which is beneficial to the reaction. The activity of the enzyme above or below the optimal temperature may affect the rate of the enzymatic reaction. When the temperature is too high, the enzyme may be inactive. So the temperature range was set to 25–45°C for 4 h. After HPLC analysis for Homoglutamate, the optimum temperature for mutant enzyme TiPDH was found 40°C as compared to the wild-type TiPDH as shown in Fig 2b.

Even at 45°C, there is no great effect on the activity of the TiPDH enzyme but only the conversion rate was low which shows that TiPDH has low catalytic efficiency for the non-natural substrate 2-ketoadipic acid.

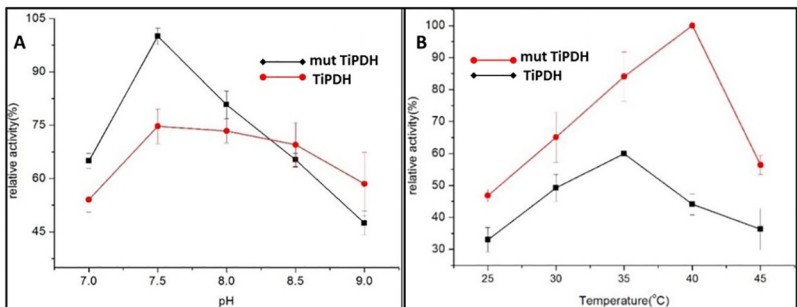

**Fig 2.** Catalytic activity of wild type and mutant TiPDH to convert 2-ketoadipic acid into homoglutamate (A) Activity of TiPDH (wild and mutant types) at different pH conditions (B) Activity of TiPDH (wild and mutant types) at different temperatures.

## Effect of adding protective agent on enzyme stability

The low conversion efficacy of transaminases reaction is due to the instability and inactivation of the mutant TiPDH enzyme. There are three leading methods to increase the stability of the enzyme, which are enzyme encapsulation, chemical modification, and immobilization [38]. In this experiment shielding mediators like PMSF, sucrose, and trehalose were selected as a protectant agent. The addition of PMSF verifies whether the enzyme is hydrolyzed during the reaction while the trehalose and sucrose play a significant part in maintaining the stability of the enzyme. The hydrogen bonds are generally believed to stabilize protein structures while helping proteins to fold. As trehalose contains a large number of hydroxyl groups, it can form hydrogen bonds with water molecules to contribute to protein stability [38, 39].

## PMSF effect on enzyme stability

To stipulate the consequence of PMSF on enzymes two conditions were set. After incubating for 2h at the optimum temperature, then rapidly placed on ice to cool down, and the enzyme activity was measured while in the second condition PMSF (protease inhibitor) was added to the final concentration of 2 mM at the same temperature for 2h. The results shown in S1a Fig indicate that the residual enzyme activity decreased after the addition of PMSF. The reason may be that after the addition of PMSF in the purification process, it binds to the serine residue in the active site of the enzyme deactivating the serine hydroxyl group through an esterification process.

## Effect of carbohydrates on enzymes

The enzyme protecting agents the trehalose and sucrose were added in a final concentration of 2 mM, 6 mM, and 10 mM enzyme solution in as shown in S1b Fig. The sucrose at a concentration of 2 mM showed a better effect whereas the trehalose at a concentration of 6 mM presents higher residual activity as compared to the blank sample after incubation at 40°C for 2 h. Then triplicate samples were set to measure the initial enzyme activities, two samples of sucrose with a final concentration of 2 mM and trehalose with 6 mM were added to the enzyme system and the third one was taken as control without trehalose and sucrose. The cells were incubated at 40°C and enzyme activity was determined after 8 and 12 h. The results indicate that trehalose and sucrose have protective effects on the enzyme as compared to the blank control one. After incubating for 12 h, the residual enzyme activity for trehalose was 80%, and the sucrose was 70% whereas the blank one was reduced to 40% shown in Fig 3a.

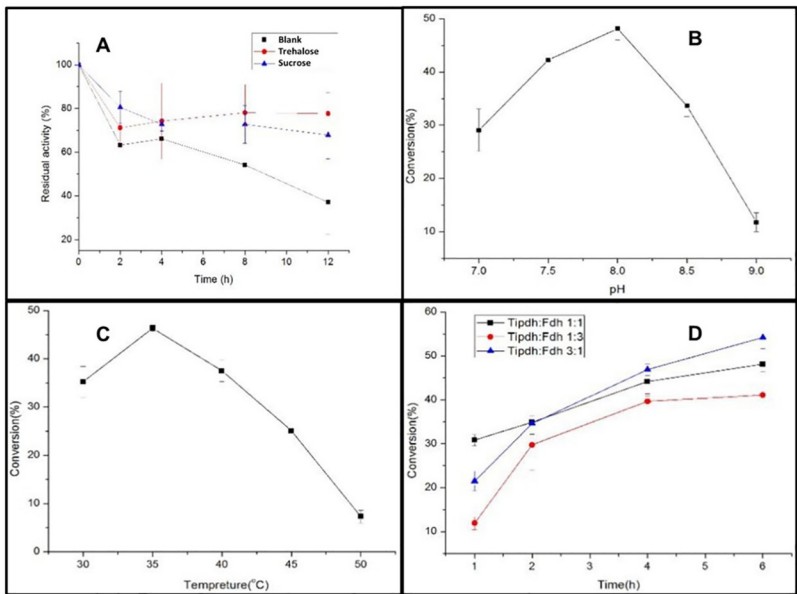

**Fig 3.** (A) Effect of Trehalose and Sucrose on enzyme TiPDH demonstrate the protective effects in the stability of the enzyme as compared to the blank control one show 40% (B) Effect of pH on the circulatory system after some time and shows a high rate of conversion at pH 8 (C) Effect of temperature on circulatory on the system and shows about 47% conversion rate of the products (D) Effect of TiPDH and FDH on the circulatory system at different ratio of TiPDH and FDH.

When the enzyme was heated, the intermediate water molecules compete with the groups on the protein molecules to form hydrogen bonds, which cause damage to the protein structure. Although enzyme protectants can also form hydrogen bonds but have less structural damage to proteins. It is because that trehalose and sucrose have a greater hydrated volume and have been ascribed to tougher and more extensive hydrogen bonding between their hydroxyl groups and water molecules. As a result, protein denaturation in such a situation would need superfluous energy to accommodate its enlarged surface area [39]. Previously, different types of osmolytes such as sucrose, sorbitol and proline were used which improved thermostability of enzyme activity, indicting greater protective effect of osmolytes as compared to our results [40].

## NADH regeneration

As NADH is a more expensive chemical and the transaminase of this experiment TiPDH consumes it during the reaction so providing them in stoichiometric amounts in the reaction system is not economically viable. For this, we introduced a coenzyme circulating system choosing formate dehydrogenase (FDH) from *Candida boidinii* as a cofactor regenerative enzyme for our system. The formate dehydrogenase is used to catalyze the oxidation of formate to carbon dioxide reducing the biological cofactor $NAD^+$ by formate ions to regenerate NADH necessary for a further round of catalysis and $CO_2$ that can be removed from the system easily. The best condition using the FDH enzyme in the reaction system was 200 μL Tris-NH4CI buffer (pH = 7–9), containing 1 mM $NAD^+$, 1g/L 2-ketoadipate, 10 mM ammonium formate, 20 μL of mut-TiPDH and FDH, the reaction was set to 35˚C. The result clarifies that the introduction of the FDH and cofactor $NAD^+$ cycle at pH 8 system can function normally and the conversion rate is 48%. But as the pH reached 9 the conversion rate dropped rapidly as

shown in Fig 3b, possibly due to the pH being too high, causing the enzyme to be reversible or irreversibly inactivated.

## Effect of temperature on the circulation system

To study the effect of temperature the reaction system was designated at the optimum pH containing Tris-NH₄CI buffer (pH = 8), and the reaction was carried out at 30˚C, 35˚C, 40˚C, 45˚C, and 50˚C, respectively. The conversion rate of homoglutamic acid was 47% after HPLC analysis at temperature 35˚C after 6h as shown in Fig 3c.

When the reaction temperature reached 50˚C, the conversion rate rapidly decreased because at high temperatures the enzyme was inactive.

## An optimal ratio of enzymes in the circulatory system

In the circulation system, 2-ketoadipate is catalyzed by mutant TiPDH to form homoglutamic acid while formate dehydrogenase catalyzes the formation of $CO_2$ by formic acid. The enzyme activity and reaction rate in the two reactions are different. Therefore, the conversion rate of homoglutamate is required. To reach the maximum level, the two enzymes mutant TiPDH and FDH were selected in a ratio of 1:3, 1:1, and 3:1 at the optimum temperature and pH respectively at the intervals of 1h, 2h, 4h, and 6h. The yield was detected by HPLC and the reaction was stopped by adding 20% hydrochloric acid.

The results illustrated the ratio of the conversion of TiPDH and FDH at 1:1 was 30%, at 3:1, and 1:3 were 22% and 12% respectively after 1 h of the reaction as shown in Fig 3d.

With the extension of time, the conversion rate of 1:3 TiPDH and FDH rapidly increased and reached the highest level of 53% in 6h whereas, the TiPDH and FDH 1:3 increased slowly and the conversion rate was 40% after 6h. The reason may be that the reaction rate of transaminase TiPDH is lower than that of formate dehydrogenase (FDH) and the reaction rate further decrease when the amount of transaminase enzyme decreased.

## Structural stability and residual flexibility analysis of the wild type and mutant complexes

Dynamics stability of both the complexes was evaluated by calculated the Root mean square deviation (RMSD). A 20ns trajectory analysis revealed that the mutant structure remained more stable than the wild type. In case of the wild type the RMSD remained higher initially until 10ns. However, after 10ns the RMSD decreased and no major convergence was observed. The average RMSD for the wild type was reported to be 1.5Å. In case of the mutant type the RMSD remained stable and no convergence was reported. The average RMSD for the mutant was reported to be 1.0 Å. RMSDs of the wild and mutant is given in Fig 4a.

To determine the residual flexibility of each systems Root mean square fluctuation (RMSF) was calculated. It can be seen that both the systems showed similar pattern of flexibility except in some regions where the wild type possess higher flexibility. These results significantly suggest that the mutants favours the binding of the substrate than the wild type and remained more stable during the simulation. RMSDs of the wild and mutant is given in Fig 4b.

To further understand the impact of the double mutant on the binding of the substrate, Gibbs free energy was calculated. Each energy term such as Van der Waal, electrostatic and the total binding energy was calculated. As given in table X the double arginine substitution favors the binding of the substrate than the wild type. The total binding energy for wild type was reported to be -34.1124 kcal/mol while for the mutant complex it was reported to be -43.6183kcal/mol. The other energy terms are given in Table 2.

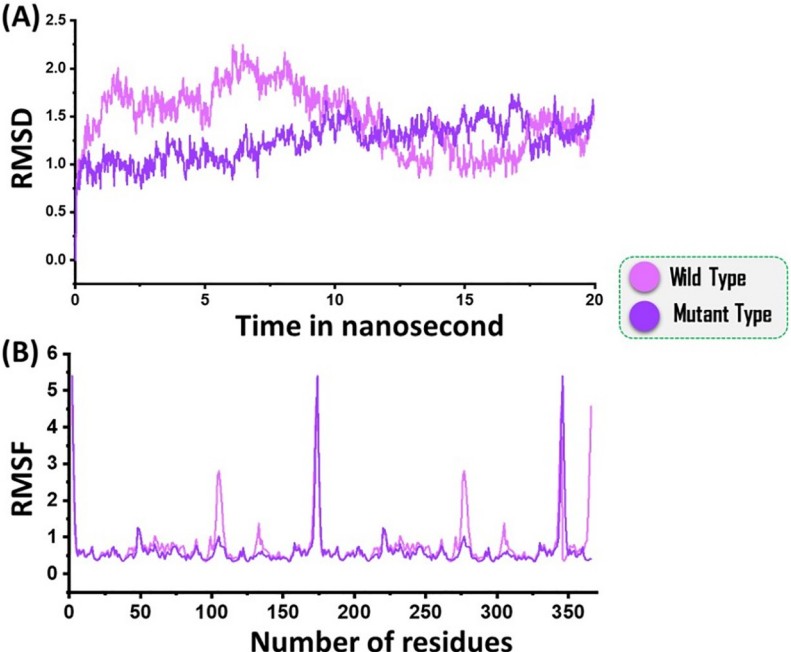

**Fig 4.** (A) showing the RMSDs of the both complexes, (B) showing the RMSFs of the complexes. Each complex is colored differently.

## Discussion

In this work, we met with some challenges: first the genetically stable framework of the host micro-organism to grow at minimal growth conditions. To overcome this problem, we choose *Escherichia coli* as a model organism due to simple genetic manipulation, clear genetic background, and one of the prominent microorganisms used in metabolic engineering [41]. Its metabolism and regulation is well characterized and has a variety of genetic tools [42] like Keio knockout collection and MAGE [43], synthetic biology tools like promoters, well-characterized regulators and ribosome-binding sites [43], and systems biology tools like genome-scale models [44]. The second challenge was the use of cofactor NADH which is an expensive chemical while using a stoichiometric amount in a chemical reaction. We used FDH formate dehydrogenase from *Candida boidinii* in our system as a cofactor regenerative enzyme. The FDH catalyzes the oxidation of formate to $CO_2$ with parallel reduction of biological cofactor NAD by formate ions to regenerate NADH [45, 46]. The third challenge was the stability of our mutant phenylalanine dehydrogenase enzyme (TiPDH), so we chose shielding mediators like PMSF, sucrose and trehalose as protectant stabilizers.

In the current study, we engineered the phenylalanine dehydrogenase enzyme (TiPDH) to obtain the high yield of novel non-proteinogenic homoglutamate. For this, we built the 3D homology model of TiPDH through the BLAST tool and MODELLER 9.12 as shown in Fig 5a

**Table 2. The binding free energy of the wild and mutant complexes.** All the energies are given in kcal/mol.

| System | vdW | Electrostatic | SASA | Total Binding |
|---|---|---|---|---|
| Wild Type | -20.55 | -08.18 | -2.3672 | -34.1124 |
| Mutant Type | -23.84 | -15.65 | -4.8472 | -43.6183 |

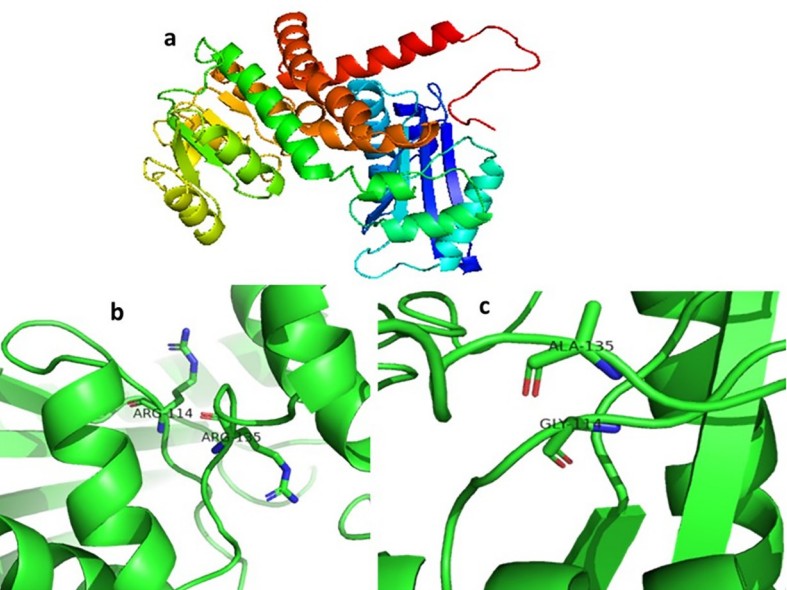

**Fig 5.** (a) Predicted structure of TiPDH through homology modeling with 1C1D (b) Docked pose of mutant TiPDH in which the Arg114 and Arg135 best fitted in the pocket. (c) Docked pose of wild type TiPDH.

and then analyzed its binding pocket through docking simulations. We have selected the enzyme structure from *Rhodococcus sp* (PDB code: 1C1D) as a template because the structure and catalytic mechanism have already been reported and provide a key basis for important amino acid residues for mutagenesis. The predicted docked pose of the TiPDH revealed that the α-Ketoadipic acid used in this study showed a promising interaction of the TiPDH enzyme. The best docking geometries of α-Ketoadipic acid and L-Phenylalanine with the most favorable binding affinities are illustrated in Fig 6a and S3 Table.

The TiPDH contains two domains separated by a deep cleft. The residues 13–133 comprise N terminal containing three helices motifs and six β-stranded sheets. The residues 200–353 forming C-terminal that comprise five β-stranded sheets core surrounded by seven α-helices and one $3_{10}$ helix shown in S2 Fig. The residues 78G, 79G, 80G, and 81K motif are conserved in various dehydrogenases. The lysine 81K in this glycine-rich region participates in the catalysis forming the Schiff base with α-ketoacids. The D69 and N264 are conserved and formed hydrogen bonding with the carboxylic group of the substrate. Besides these, the residues R45, D116, and T151 are also conserved and are responsible for the binding of cofactor NAD (P) $^+$. The tertiary structure of the coenzyme binding domain of NAD (P) $^+$ dehydrogenase contains four-stranded parallel β sheets and an α-helix with an essentially indistinguishable arrangement and is highly conserved [17].

We observed that the functional moieties of these molecules play an important role in the interactions. Molecular insights based on simulation revealed the reason behind the higher activity of α-Ketoadipic acid. The ligand L-Phenylalanine showed important interactions with the crucial moieties in the substrate-binding pocket as shown in Fig 6b.

The Asn264 interacts with the carboxyl moiety through hydrogen bonding with a distance of 2.97Å, similarly, also Thr115 interacts with an amino group formed hydrogen bonding with a bond length of 2.98Å. Besides, these favorable contacts there are also other sensible hydrophobic interactions observed from the surrounding pocket, like Leu41, Gly42, Gly43, Met66, Lys69, Gly114, and Leu294. Furthermore, it should be noted that the terminal carboxylic

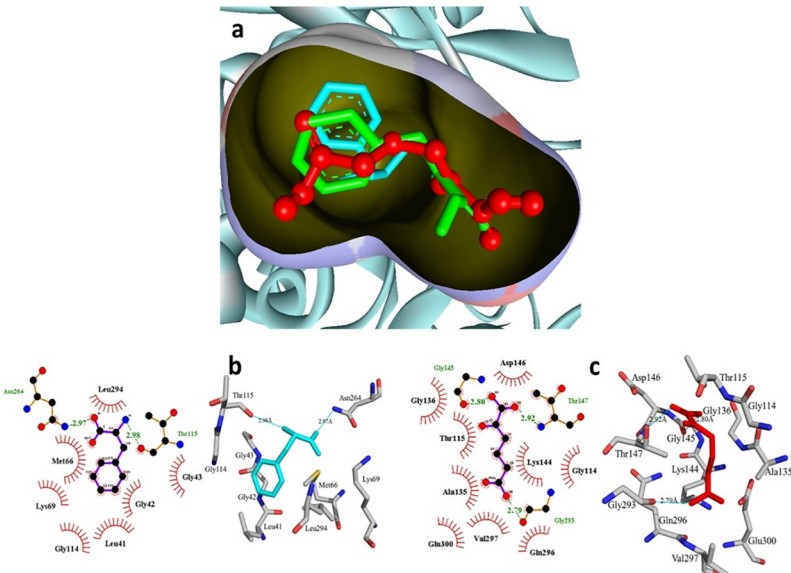

**Fig 6.** (a) Illustration of the predicted docked poses of wild type TiPDH against the α-Ketoadipic acid (red color ball and stick) and the phenylalanine (green color sticks) while the ligand shown by the cyan color sticks (b) The 2D and 3D representation of the natural substrate (Phenylalanine) in the binding pocket of wild type TiPDH enzyme (c) The 2D and 3D representation of docked α-Ketoadipic acid (our substrate) in the binding pocket of wild type TiPDH enzyme.

functional moieties of α-Ketoadipic acid significantly interact through three hydrogen bonding with the important residues such as Gly145 (2.80 Å), Thr147 (2.92 Å) and Gly293 (2.79 Å). It should be noteworthy, that there are around nine hydrophobic contacts that were detected in the binding pocket of α-Ketoadipic acid including Gly114, Thr115, Ala135, Gly136, Lys144, Asp146, Gln296, Val297, and Glu300 Fig 6c.

Phenylalanine dehydrogenases from different enzymes have shown substantial resemblance particularly in the region between D58-E130. The region from D58 to E130 contains a 78G-G-G-81K domain, which is common in different aminoacid dehydrogenases and probably constitutes the catalytic domain [17].

Based on computational insight to the structure, we proposed that the site-specific directed mutagenesis of the two important residues in the predicted binding pocket of α-Ketoadipic acid Gly114/Arg114 and Ala135/Arg135 led to the major reason showing the stronger affinity and was best fitted inside the binding pocket Fig 5b and 5c, and resulted in the best activity.

It was strongly supported through significant two hydrogen-bonding interactions by mutated Arg114 (2.93 Å and 2.73 Å) with the α-Ketoadipic acid while the wild type Gly114 only showed hydrophobic interactions. Additionally, in the wild type Ala135 formed only hydrophobic interactions while the mutated Arg135 formed one hydrogen bonding with the α-Ketoadipic acid Fig 7a and 7b.

Furthermore, the structural interactions of both α-Ketoadipic acid and L-Phenylalanine showed the most favorable electrostatic interactions with the active site of the enzyme but the shape and steric bulk will be the limiting factor in the biological conversion of α-Ketoadipic acid and L-Phenylalanine. The L-Phenylalanine was found to have a lesser affinity because its molecular size and weak interactions could create a barrier towards its binding inside the active site of the enzyme.

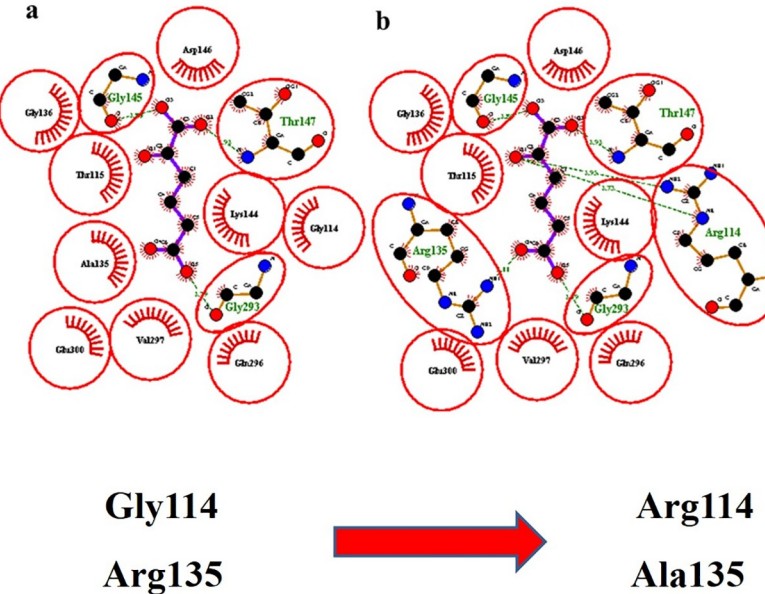

**Gly114**

**Arg135**

**Arg114**

**Ala135**

**Fig 7. The 2D illustration of the docked pose of α-Ketoadipic acid in the binding pocket of the TiPDH enzyme.** (a) The pose of α ketoadipic acid with wild type TiPDH (b) Pose of α-Ketoadipic acid with the mutated TiPDH.

## Conclusions

In the summary, two single Ala135Arg, Gly114Arg, and one double Gly114Arg/Ala135Arg mutation in TiPDH were carried out for the amination of ketoadipic acid. The catalytic efficiency Km/Kcat of the mutant Ala135Arg showed a 3-fold increase towards the ketoadipic acid as compared to the other mutant Gly114Arg, a double mutant Gly114Arg/Ala135Arg, and wild type TiPDH. Besides, the formate dehydrogenase (FDH) in the system was applied as a cofactor regenerative enzyme to make the system economically viable. Some of the shielding mediators like PMSF, sucrose, and trehalose were chosen as a protectant in the system to maintain the stability of the enzyme. Our study contributes to the expansion of the synthetic scope of phenylalanine dehydrogenases catalyzed asymmetric reductive amination and unfolding the possibility of biosynthesis of other non-proteinogenic amino acids that might be valuable pharmaceutical intermediaries.

## Supporting information

**S1 Fig.** (**a**) Effect of PMSF on substrate decreased the residual enzyme activity. (**b**) Effect of different concentration of Trehalose and Sucrose showed protective effects in the stability of the enzyme.
(DOCX)

**S2 Fig. TiPDH sequence containing thirteen α helices and twelve β strands sheets and some conserved regions.** The red underlined K69, K81, and N264 are conserved residues and are involved in the binding of substrate. Whereas the highlighted residues encircled in the red boxes are involved in the binding of cofactors.
(DOCX)

**S1 Table. Plasmids used in this work.**
(DOCX)

**S2 Table. Primers for wild type and mutant enzymes used in this study.**
(DOCX)

**S3 Table. Molecular docking statistics of α-Ketoadipic acid and Phenylalanine with the binding pocket of TiPDH.**
(DOCX)

## Acknowledgments

All the authors are grateful to Dr. Muhammad Asim Department of Biology, The University of Haripur for his constructive suggestions and help throughout manuscript writing.

### Ethical approval

This article does not contain any studies with human participants or animals performed by any of the authors.

## Author Contributions

**Conceptualization:** Shafiq Ur Rehman, Xinxiao Sun, Qipeng Yuan.

**Data curation:** Bashir Ahmad, Azizullah Azizullah.

**Formal analysis:** Muhammad Tariq, Azizullah Azizullah, Shafiq Ur Rehman, Muhammad Faheem, Xinxiao Sun.

**Investigation:** Muhammad Tariq.

**Methodology:** Muhammad Tariq.

**Project administration:** Xinxiao Sun.

**Software:** Qipeng Yuan.

**Supervision:** Muhammad Israr, Muslim Raza, Qipeng Yuan.

**Validation:** Bashir Ahmad, Qipeng Yuan.

**Visualization:** Bashir Ahmad, Qipeng Yuan.

**Writing – original draft:** Muhammad Israr, Muslim Raza, Azizullah Azizullah, Shafiq Ur Rehman, Qipeng Yuan.

**Writing – review & editing:** Muhammad Tariq, Muhammad Israr, Muslim Raza, Bashir Ahmad, Azizullah Azizullah, Shafiq Ur Rehman, Xinxiao Sun, Qipeng Yuan.

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
