## [Decision Letter · Decision Letter 0]

23 Aug 2021

PONE-D-21-21697

Engineering of Phenylalanine Dehydrogenase from Thermoactinomyces Intermedius for the Production of a novel Homoglutamate

PLOS ONE

Dear Dr. Israr,

Thank you for submitting your manuscript to PLOS ONE. After careful consideration, we feel that it has merit but does not fully meet PLOS ONE’s publication criteria as it currently stands. Therefore, we invite you to submit a revised version of the manuscript that addresses the points raised during the review process.

Please address as fully as possible the comments of both reviewers, especially the detailed and constructive comment of Reviewer 2.

We look forward to receiving your revised manuscript.

Kind regards,

Israel Silman

Academic Editor

PLOS ONE

Journal Requirements:

Reviewers' comments:

Reviewer's Responses to Questions

**Comments to the Author**

1. Is the manuscript technically sound, and do the data support the conclusions?

Reviewer #1: Yes

Reviewer #2: Partly

2. Has the statistical analysis been performed appropriately and rigorously? 

Reviewer #1: No

Reviewer #2: N/A

3. Have the authors made all data underlying the findings in their manuscript fully available?

Reviewer #1: Yes

Reviewer #2: No

4. Is the manuscript presented in an intelligible fashion and written in standard English?

Reviewer #1: Yes

Reviewer #2: Yes

5. Review Comments to the Author

Reviewer #1: 1. The whole data re suffering from lack of sufficient statistical analysis, it should be added more emphasized on kinetic data.

2. I am not sure about kcat/km and km/kcat difference? i presume they are the same and should be corrected in the whole manuscript.

3. Is there any data regarding specificity of native and mutant enzymes towards Tyrosine compared phenylalanine. Based on whatever reported in the literature it should be at least be discussed, as reported in:

Archives of Biochemistry and Biophysics

Volume 635, 1 December 2017, Pages 44-51.

4. Kcat/Km ratio also should be shown in table.

5. in the literature review , i presume similar papers on the effect of enzyme stabilizers could be used:

-International Journal of Biological Macromolecules, Volume 43, Issue 2, 15 August 2008, Pages 187-191.

-Journal of Molecular Catalysis B: Enzymatic

Volume 62, Issue 2, February 2010, Pages 127-132

Reviewer #2: General Comments

The work “Engineering of phenylalanine dehydrogenase…” PONE D 21 21697, by Tariq et al. is dedicated to the design of an enzymatic system to obtain the non-proteinogenic amino acid homoglutamate. This compound may have medical interest and is usually known as alpha aminoadipic acid (AAA). In the Introduction the authors describe several AAA biosynthetic methods that may be used to obtain this compound; however, they do not focus the article on the use of the naturally producing AAA biosynthetic enzymes, the names of which are not cited. These enzymes are present in actinobacteria or fungi and are well known to form AAA as precursor of the beta-lactam antibiotics. Instead, the authors choose to study the phenylalanine dehydrogenase of Thermoactinomyces intermedius an enzyme that, as the authors indicate in line 336, lacks efficiency for the amination of alpha ketoadipic acid, the compound used by the authors as substrate. Therefore, it is surprising that the authors choose to modify the phenylalanine DH which belong to type I (aromatic) aminotransferases to develop this project. The experimental part of the work dedicated to the modification of the T intermedius PDH is largely correct although there are some obscure points in the quantification method as indicated below. The authors also studied the regeneration of NADH cofactor for the TiPDH by using the well known formate DH (decarboxylase) that produces NADH. The discussion is entirely focused on the modification of the protein that the authors have performed but they fail to discuss the main aim of this project which is the microbial production of AAA by microbial biosynthetic methods which are easy and highly productive.

Specific comments

1. In the abstract the authors refer to the compound of interest as homoglutamate; however, this compound is usually named 2-aminoadipic acid or alpha aminoadipic acid. Please, use at least in the abstract the two names, so that the readers can identify the compound that is being investigated as you have done in the first sentence of Introduction.

2. In lines 55 the authors indicate that homoglutamate is a putative acyclic precursor of penicillins an cephalosporins. This word “putative” in this sentence is misleading; it is very well stablished in numerous articles that alpha aminoadipate (AAA) is a real precursor of penicillins and of cephalosporin C in P. chrysogenum and C. acremonium, respectively. Please clarify this point eliminating the word “putative” and providing more recent references. Also provide information on the enzymes producing AAA in actinobacteria and fungi.

3. In lines 69-70 it is indicated that there is no native pathway for the biosynthesis of alpha aminoadipate. This sentence is wrong; it is well known that there are pathways for AAA which are different in distinct microorganisms as actinobacteria and filamentous fungi. Please modify the sentence

4. In line 73 that authors state that AAA is an intermediate in the catabolism of lysine in mammals and in beta-lactam producing actinomycetes (Esmahan et al 1994). The work of Esmahan was made in filamentous fungi not in actinomycetes. This is an important microbiological mistake since actinomycetes are bacteria.

5. Lines 73-76. The authors say “However, this non proteinogenic amino can be…”. Please correct amino to amino acid. The term “however” is not adequate since it means that there is a contradiction with the previous sentence, which is not the case. Change the term “however” to “in addition…”. Please correct also the reference of Luengo et al in line 76 and in the references list.

6. Lines 93-95 the �-aminotransferase reaction which convert the alpha ketoadipic acid is into alpha amino adipic acid is probably the best approach to obtain unexpensive AAA- The mechanism of action of the � aminotransferase is still poorly known but could be explored

7. In the second part of the introduction the author propos to use the phenylalanine dehydrogenase from Thermoctinomyces to produce AAA from alpha cetoadipic. This is a surprising approach since the substrate of phenylalanine DH has a 9 carbons aromatic substrate that is deaminate to phenylpyruvate; this rection biochemically is quite different from that converting alpha-ketoadipate in AAA. The authors should clarify why they choose to use this reaction for the conversion but they do not approach the modification of enzymes related to the bacterial or fungal formation of AAA. The authors should give a clear study of the relative efficiency of AAA formation by the phenylalanine DH in relation to the natural enzymes forming AAA.

8. Line 118 and in the Results section the authors indicate that the affinity of the TiPDH for transamination of the 2-cetoadipate substrate is insufficient and therefore they try to improve the production by mutagenesis of two amino acids of the putative active center. This observation indicates that this is not the best starting point for the development of the high production of AAA.

9. In Materials and Methods the authors describe the quantification of AAA formation by HPLC but, as the authors indicate previously, homoglutamate (AAA) has not good UV absortion and the method utilized is not clear. In all classical studies amino acids are measured after derivatization with OPA or other UV emitting agent. Please indicate which is the method used in this article.

10. In the Discussion section the authors focus the discussion in the structure of the TiPDH and but they do not compare the active center of this enzyme with those of other alpha aminoadipic forming enzymes.

11. The authors in line 142 give the accession number of the nucleotides sequence of the TiPDH encoding gene, D00631. However, since in the article the authors constantly refer to the TiPDH protein the accession number of the protein should be easily accessible in the text. We found that the amino acids sequence shown in figure S2 does not corresponds to the sequences published by Ohshima et al or Tanaka et al. Is a different protein?. Please clarify.

12. In line 87 the authors refer to the work of Perez Llarena refering to an enzymatic reaction. This work is purely a description of gene sequences. The real work of purification and characterization of the enzyme in that of “de la Fuente et al., 1997”

6. PLOS authors have the option to publish the peer review history of their article (what does this mean?). If published, this will include your full peer review and any attached files.

Reviewer #1: No

Reviewer #2: No

---

## [Author Response · Author response to Decision Letter 0]

20 Dec 2021

Dear editor

We would like thanks to you and reviewers for their precious time to review our manuscript in deep and critically and for suggesting corrections that improved the quality of our manuscript. We are pleased to submit the revised manuscript entitled “Engineering of Phenylalanine Dehydrogenase from Thermoactinomyces Intermedius for the Production of a novel Homoglutamate” for consideration in PLOS ONE. On the following pages, you will find our response to the reviewer comments. The reviewer comments and suggestions are highlighted with red color and responses to each point with normal text. On behalf of my co-authors, I thank you for your consideration of this resubmission. We appreciate your time and look forward to your response. 

Sincerely, 

Dr. Muhammad Israr PhD, Postdoc (Corresponding Author)

Associate Professor, Department of Biology 

The University of Haripur, KPK, Pakistan 

Reviewers comments and Responses 

Reviewer 

Comment 1: In the abstract, the authors refer to the compound of interest as homoglutamate; however, this compound is usually named 2-aminoadipic acid or alpha aminoadipic acid. Please, use at least in the abstract the two names, so that the readers can identify the compound that is being investigated as you have done in the first sentence of the Introduction.

Response 1: We are grateful to the reviewer for taking his precious time to deeply review our manuscript which enhanced its quality.

Following the reviewer's suggestion, we have added two names of 2-aminoadipic acid in the Abstract section.

Comment 2: 

Abstract: In line 55 the authors indicate that homoglutamate is a putative acyclic precursor of penicillins and cephalosporins. The word “putative” in this sentence is misleading; it is very well established in numerous articles that alpha aminoadipate (AAA) is a real precursor of penicillins and of cephalosporin C in P. chrysogenum and C. acremonium, respectively. Please clarify this point by eliminating the word “putative” and providing more recent references. Also, provide information on the enzymes producing AAA in actinobacteria and fungi.

Response 2: Thanks for this comment and suggestions. Yes, we are agreeing with the reviewer, the word “Putative” was replaced with “Well-known” as suggested by the reviewer and highlighted with red color in the track changes. 

Comment 3: 

In lines 69-70 it is indicated that there is no native pathway for the biosynthesis of alpha aminoadipate. This sentence is wrong; it is well known that there are pathways for AAA which are different in distinct microorganisms as actinobacteria and filamentous fungi. Please modify the sentence

Response 3: We are obliged to the reviewer for this comment and agree with his suggestion for correction. The sentence is modified and highlighted with red color in the track changes.

Comment 4:

In line 73 the authors state that AAA is an intermediate in the catabolism of lysine in mammals and beta-lactam producing actinomycetes (Esmahan et al 1994). The work of Esmahan was made in filamentous fungi, not in actinomycetes. This is an important microbiological mistake since actinomycetes are bacteria.

Response 4: We are grateful to the reviewer for pointing out this mistake. In the track changes, the sentence is corrected and the word “actinomycetes” is replaced with “Filamentous fungi” which is the work of Esmahan et al 1994. 

Comment 5: 

Lines 73-76. The authors say “However, this nonproteinogenic amino can be…”. Please correct amino to an amino acid. The term “however” is not adequate since it means that there is a contradiction with the previous sentence, which is not the case. Change the term “however” to “in addition…”. Please correct also the reference of Luengo et al in line 76 and the references list. 

Response 5: We are grateful to the reviewer for this comment. The sentence and reference are modified accordingly and highlighted with red color in track changes.

Comment 6: 

Lines 93-95 the ω-aminotransferase reaction which converts the alpha ketoadipic acid is into alpha-amino adipic acid is probably the best approach to obtain unexpansive AAA- The mechanism of action of the ω- aminotransferase is still poorly known but could be explored

Response 6: We are obliged to the reviewer for this comment. In some organisms (bacteria/fungi) the L-Lysine is converted into α-aminoadipic acid by two different pathways, in one pathway, lysine 2-ketoglutarate reductase results in the formation of saccharopine from lysine by extracting one amino group to be used as a nitrogen source. While in the second pathway the ω-aminotransferase converts D-lysine or L-lysine into 2-aminoadipic semialdehyde and 2-aminoadipic acid (homoglutamic acid). Both enzymes are prompted by L-lysine which is negatively regulated by NH4+ ions. The ω-aminotransferase from different species uses 2-ketoacids as amino group acceptors. The most common ones are 2-ketoadipate and 2-ketoglutarate resulting directly in the development of 2-aminoadipic acid and glutamic acid. We are agreed with the reviewer that ω--aminotransferase reaction which converts the alpha ketoadipic acid is into alpha-amino adipic acid is probably the best approach to obtain unexpansive AAA- The mechanism of action of the ω- aminotransferase is still poorly known but it could be explored.

Comments 7:

In the second part of the introduction, the author proposes to use the phenylalanine dehydrogenase from Thermoactinomyces to produce AAA from alpha ketoadipic. This is a surprising approach since the substrate of phenylalanine DH has 9 carbons aromatic substrate that is delaminated to phenylpyruvate; this reaction biochemically is quite different from that converting alpha-ketoadipate in AAA. The authors should clarify why they choose to use this reaction for the conversion but they do not approach the modification of enzymes related to the bacterial or fungal formation of AAA. The authors should give a clear study of the relative efficiency of AAA formation by the phenylalanine DH in relation to the natural enzymes forming AAA.

Response 7: We are thankful to the reviewer for this comment. We have chosen reductive amination reaction strategies for the production of homoglutamate by proposing the phenylalanine dehydrogenase from Thermoactinomyces using alpha ketoadipic which is an interesting route for the conversion of 2-ketoadipic acid catalyzed by NADH-dependent phenylalanine-dehydrogenase. The natural TiPDH enzyme affinity towards the transamination of 2-ketoadipic acid was previously found insufficient, therefore, we have chosen some amino acids sites in TiPDH and mutated which enhanced that catalytic activity for the production of AAA.

Comment 8:

Line 118 and in the Results section the authors indicate that the affinity of the TiPDH for transamination of the 2-cetoadipate substrate is insufficient and therefore they try to improve the production by mutagenesis of two amino acids of the putative active center. This observation indicates that this is not the best starting point for the development of the high production of AAA.

Response 8: We are obliged to the reviewer for this comment. In the introduction (line 111-116) and result section, we have modified the sentence in the track changes and highlighted it with red color according to the reviewer's suggestion. 

Comment 9:

In Materials and Methods, the authors describe the quantification of AAA formation by HPLC but, as the authors indicate previously, homoglutamate (AAA) has not had good UV absorption and the method utilized is not clear. In all classical studies, amino acids are measured after derivatization with OPA or other UV emitting agents. Please indicate which is method used in this article.

Response 9: The authors are grateful to the reviewer for this comment. The method used in the article is explained in reference 36.

Comment 10:

In the Discussion section, the authors focus the discussion on the structure of the TiPDH and but they do not compare the active center of this enzyme with those of other alpha aminoadipic forming enzymes.

Response 10: We are obliged to the reviewer for raising this point. We have included some sentences of the revised manuscript (discussion section lines 454-456) which compared the active site domain of different alpha aminoadipic forming enzymes. We are the first to mutate the active site residues of TiPDH and no literature is available for comparing the active site of TiPDH with other alpha aminoadipic forming enzymes.

Comment 11:

The authors in line 142 give the accession number of the nucleotides sequence of the TiPDH encoding gene, D00631. However, since in the article, the authors constantly refer to the TiPDH protein the accession number of the protein should be easily accessible in the text. We found that the amino acids sequence shown in figure S2 does not correspond to the sequences published by Ohshima et al or Tanaka et al. Is it a different protein? Please clarify.

Response 11: We appreciate the reviewer for reviewing our article in deep. The Accession Number D00631 is for TiPDH which we used throughout the manuscript that can easily be assessable and can be recognized. The amino acid sequences in the S1 Figure are that of TiPDH. The protein sequences of TiPDH and those published by Oshima T and Takada H are the same proteins but from a different organism that has a resemblance in amino acid residues at the active site domain.

Comment 12:

In line 87 the authors refer to the work of Perez Llarena referring to an enzymatic reaction. This work is purely a description of gene sequences. The real work of purification and characterization of the enzyme in that of “de la Fuente et al., 1997”

Response 12: We are thankful to the reviewer for this comment. We are agreed with the reviewer that the reference of Perez Llarena is not for PDH enzymatic reaction, so in the revised manuscript we removed the Perez Llarena reference and replaced it with De la Fuente et al.,

---

## [Decision Letter · Decision Letter 1]

20 Jan 2022

PONE-D-21-21697R1Engineering of Phenylalanine Dehydrogenase from Thermoactinomyces Intermedius for the Production of a novel HomoglutamatePLOS ONE

Dear Dr. Israr,

Thank you for submitting your manuscript to PLOS ONE. After careful consideration, we feel that it has merit but does not fully meet PLOS ONE’s publication criteria as it currently stands. Therefore, we invite you to submit a revised version of the manuscript that addresses the points raised during the review process. In your revised manuscript please address in full the minor comments made by Reviewer 1.

We look forward to receiving your revised manuscript.

Kind regards,

Israel Silman

Academic Editor

PLOS ONE

Reviewers' comments:

Reviewer's Responses to Questions

**Comments to the Author**

1. If the authors have adequately addressed your comments raised in a previous round of review and you feel that this manuscript is now acceptable for publication, you may indicate that here to bypass the “Comments to the Author” section, enter your conflict of interest statement in the “Confidential to Editor” section, and submit your "Accept" recommendation.

Reviewer #1: All comments have been addressed

2. Is the manuscript technically sound, and do the data support the conclusions?

Reviewer #1: Yes

3. Has the statistical analysis been performed appropriately and rigorously? 

Reviewer #1: Yes

4. Have the authors made all data underlying the findings in their manuscript fully available?

Reviewer #1: Yes

5. Is the manuscript presented in an intelligible fashion and written in standard English?

Reviewer #1: Yes

6. Review Comments to the Author

Reviewer #1: 1. In Fig. 1 a and 1 b , km and vmax units should be added.

2. For the effect of sucrose and trehalose , previous results on other enzymes are stronger, please discuss:

International journal of biological macromolecules 43 (2), 187-191, 2008.

3. Other variants of phenylalanine dehydrogenase have been reported and should be used

In introduction part: Archives of biochemistry and biophysics 635, 44-51, 2017

7. PLOS authors have the option to publish the peer review history of their article (what does this mean?). If published, this will include your full peer review and any attached files.

Reviewer #1: No

---

## [Author Response · Author response to Decision Letter 1]

24 Jan 2022

Dear editor

We would like thanks to you and reviewers for their precious time to review our manuscript in deep and critically and for suggesting corrections that improved the quality of our manuscript. We are pleased to submit the revised manuscript entitled “Engineering of Phenylalanine Dehydrogenase from Thermoactinomyces Intermedius for the Production of a novel Homoglutamate” for consideration in PLOS ONE. On the following pages, you will find our response to the reviewer comments. The reviewer comments and suggestions are highlighted with red color and responses to each point with normal text. On behalf of my co-authors, I thank you for your consideration of this resubmission. We appreciate your time and look forward to your response. 

Sincerely, 

Dr. Muhammad Israr PhD, Postdoc (Corresponding Author)

Associate Professor, Department of Biology 

The University of Haripur, KPK, Pakistan 

Reviewers comments and Responses 

Reviewer 

Comment 1: In Fig. 1 a and 1 b, km and Vmax units should be added.

Response 1: We are grateful to the reviewer for pointing out this mistake. The units of Km and Vmax are added in the revised manuscript.

Comment 2: 

For the effect of sucrose and trehalose, previous results on other enzymes are stronger, please discuss: International journal of biological macromolecules 43 (2), 187-191, 2008.

Response 2: Thanks for this comment. In line 404-406 of result part, we have added a sentence with reference for comparing the effect of osmolytes such as sucrose on the thermal stability of previously reported enzyme (firefly luciferase) with our results.

Comment 3: 

Other variants of phenylalanine dehydrogenase have been reported and should be used

In introduction part: Archives of biochemistry and biophysics 635, 44-51, 2017 

Response 3: We are obliged to the reviewer for this comment. In line 97-98 of introduction part, a reference for reporting phenylalanine dehydrogenase from Bacillus badius has added accordingly.

---

## [Editor Report · Decision Letter 2]

27 Jan 2022

Engineering of Phenylalanine Dehydrogenase from Thermoactinomyces Intermedius for the Production of a novel Homoglutamate

PONE-D-21-21697R2

Dear Dr. Israr,

We’re pleased to inform you that your manuscript has been judged scientifically suitable for publication and will be formally accepted for publication once it meets all outstanding technical requirements.

Kind regards,

Israel Silman

Academic Editor

PLOS ONE
---

## [Editor Report · Acceptance letter]

24 Feb 2022

PONE-D-21-21697R2 

Engineering of Phenylalanine Dehydrogenase from *Thermoactinomyces Intermedius* for the Production of a novel Homoglutamate 

Dear Dr. Israr:

I'm pleased to inform you that your manuscript has been deemed suitable for publication in PLOS ONE. Congratulations! Your manuscript is now with our production department. 

Kind regards, 

on behalf of

Prof. Israel Silman 

Academic Editor

PLOS ONE